# COMPACT MULTIMODAL CONTEXT REPRESENATIONS USING VISUAL TOKENS

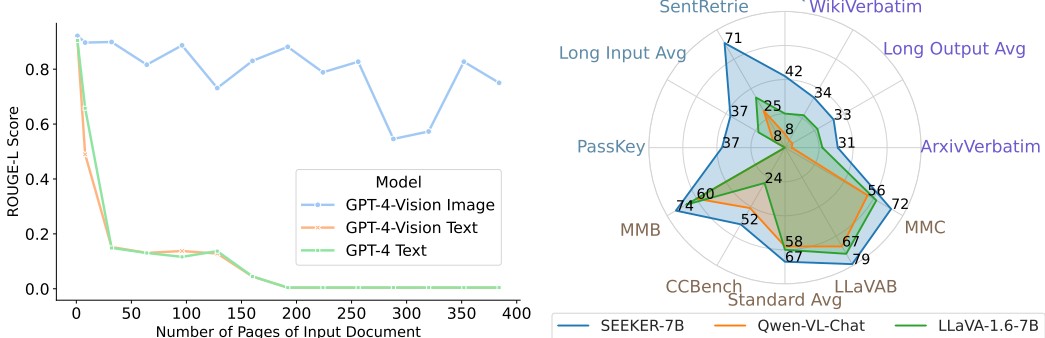

Figure 1: Left: Performance plot on First-Sentence-Retrieval task revealing compact nature of image tokens in representing long content. Right: Radar chart demonstrating the superior performance of the SEEKER (ours) model across both short and long-context multimodal tasks.

## ABSTRACT

The rapid progress in Multimodal Large Language Models (MLLMs) has significantly advanced their ability to process and understand complex visual and textual information. However, the integration of multiple images and extensive textual contexts remains a challenge due to the inherent limitation of the models' capacity to handle long input sequences efficiently. In this paper, we introduce SEEKER, a multimodal large language model designed to tackle this issue. SEEKER aims to optimize the compact encoding of long text by compressing the text sequence into the visual pixel space via images, enabling the model to handle long text within a fixed token-length budget efficiently. Our empirical experiments on six long-context multimodal tasks demonstrate that SEEKER can leverage fewer image tokens to convey the same amount of textual information compared with the OCR-based approach, and is more efficient in understanding long-form multimodal input and generating long-form textual output, outperforming all existing proprietary and open-source MLLMs by large margins.

## 1 INTRODUCTION

The success of Large Language Models (LLMs) OpenAI (2022); Touvron et al. (2023b); Bai et al. (2023a); DeepSeek-AI et al. (2024) has significantly impacted various fields, notably Multimodal Large Language Models (MLLMs) OpenAI (2023b); Liu et al. (2023c); Bai et al. (2023b); Lu et al. (2024). And there is a burgeoning interest in enhancing LLMs to handle longer context Xiong et al. (2023); Chen et al. (2024); Jin et al. (2024), for example, the recent GPT-4O OpenAI (2024) can support up to 128k tokens, paving the way to unlock many real-world applications from long-document understanding, summarization to document translation, among others.

In many applications involving long-form documents that integrate images and text, there is a significant demand for the strong long-context understanding ability of MLLMs. As shown in Figure 2, the long context in the multimodal domain falls into two main categories: 1) long-form inputs consisting of multiple text-rich images, and 2) long-form text outputs. In the first category,

| Long Image Context | Long Text Context | Long Text Generation | Long Multimodal Context |
|---|---|---|---|
| Text   **Multi-Image** | **Long Text**   Image | Text   Image | Text   **Multi Text-Rich Image** |
| Short Text Output | Short Text Output | Long Text Output | Long Text Output |

Figure 2: Long Multimodal Context Task mainly consists of two elements: 1) long image sequence and text input and 2) long text output.

multiple images increase the context length with image tokens and additional text tokens if the images are text-rich. This requires the model to efficiently integrate textual data with multiple images and reason across them. In the second category, the model must produce coherent and attentive long responses to the input context, avoiding irrelevant or hallucinated content and minimizing reliance on the model knowledge without considering the specific multimodal context.

The existing MLLMs Liu et al. (2023c;a); Lu et al. (2024) leverage pretrained LLMs Chiang et al. (2023); Touvron et al. (2023a) and inherit their advanced language understanding capabilities. Although these MLLMs demonstrate strong performance across various vision-language benchmarks Liu et al. (2024b); Yu et al. (2023), their effectiveness in long-form multimodal contexts is less explored. This issue becomes significant in tasks with very long input or output, which may exceed the context length limit (e.g., $2048$ tokens for LLaMA) and increase computational overhead.

While only a few MLLMs OpenAI (2023b); McKinzie et al. (2024) are capable of handling multiple images in the multimodal context, efficiency emerges as another critical challenge. "A picture is worth a thousand words", for human, it is more natural to fully utilize our bandwidth to process an image than words. However, this might not be the case for models. In this paper, we aim to represent information in a more compact form, enabling conveying more information within the same context length. Specifically, we investigate the "visual token representation" as an alternative to text tokens, and introduce SEEKER, an efficient method for managing long contexts within a constrained length budget. This approach allows us to process more context within a fixed token length.

As shown in Figure 3, an OCR-based approach might yield $10k$ tokens from an eight-page document for the LLM with a context limit of $8k$ tokens. While, SEEKER processes each of the eight pages as separate images, converting them into 576 tokens each. This generates a total of $4,608$ tokens for the whole document, which are then fed into the SEEKER model for reasoning and generation.

To the best of our knowledge, SEEKER is the first to address this in the long-context MLLMs by employing a compact tokenization strategy that leverages visual tokens for textual information, thus reducing the number of tokens required and enabling the processing of longer texts without additional computation overhead. SEEKER's design allows for sophisticated reasoning across multiple images. By interleaving image tokens with textual data, SEEKER can preserve context coherence and continuity across extended sequences, enabling more effective interpretation and integration of visual data in scenarios where traditional text-based models may struggle. To sum up, our main contributions are as follows:

- We present SEEKER, a novel approach to leverage the visual tokens to represent both image and text information in long documents. Our approach is more efficient than OCR text tokens, when given the same token length constraint.
- Our SEEKER supports long-context multimodal reasoning, effectively handling long-form multi-image input and generating long-form text output.
- Our instruction-tuned SEEKER model demonstrates promising results compared to the existing MLLMs on six long-context multimodal tasks.

## 2 BACKGROUND

**Multimodal Large Language Model** Recent advancements of proprietary Large Language Models, GPT-4 OpenAI (2023a), Gemini Team et al. (2023), Claude, QWen Bai et al. (2023a), and open-source ones, LLaMA Touvron et al. (2023a;b), Mistral, have shown groundbreaking applications. Their counterparts in the visual domain are followed up, including GPT-4V OpenAI (2023b), Gemini-

Figure 3: Our SEEKER surpass OCR-based model on long multimodal context tasks: 1) process multiple text-rich images naturally. 2) more compact token and fit easily in fix-context length LLM.

Vision Team et al. (2023), Claude3-Opus-VL, Qwen-VL Bai et al. (2023b), InstructBLIP Dai et al. (2023), LLaVA Liu et al. (2023d). Some work Lu et al. (2023); Wu et al. (2024) reveals the deficit of these MLLMs in multiple images reasoning, and recent models McKinzie et al. (2024); Laurençon et al. (2024); Jiang et al. (2024) improve such capabilities. Other workRust et al. (2023); Gao et al. (2024) explore to process both text and images within pixels via task-specific finetuning. However, the long-context capabilities of these MLLMs are underexplored. Our proposed SEEKER advances the long-context multimodal understanding of MLLMs from two aspects, long-form image inputs and long-form text outputs.

**Long Context Transformer** The Transformer-dominated LLMs have struggled with long context length as studied in Liu et al. (2023e). LongLLaMA Tworkowski et al. (2023), Self-Extend Jin et al. (2024) have been proposed to increase the effective context length by either fine-tuning or training-free approach based on pre-trained LLMs . When it comes to MLLMs, additional long-context issues are introduced from Vision Transformers (ViTs) Dosovitskiy et al. (2021) for image processing, and connecting with the LLMs. The concept of Dynamic Tokens Wang et al. (2021) introduces a novel approach where the allocation of computational resources is adapted dynamically, emphasizing that not all image parts equally contribute to the recognition task. Additionally, the development of the Self-slimmed Vision Transformer Zong et al. (2022) introduces a mechanism for model slimming during the inference phase, reducing computational overhead without significant loss in accuracy. In contrast, our proposed SEEKER utilizes image tokens as compact representations for image and text, alleviating the context length required for the same amount of semantic information in the language model backbone when processing multimodal content.

## 3 SEEKER: LONG-CONTEXT VISION AND LANGUAGE UNDERSTANDING

We propose SEEKER, a multimodal large language model designed to handle long-context images and texts, as depicted in Figure 3. In Section 3.1, we discuss the innovative use of image tokens to represent lengthy textual data compactly. Then we introduce long-context multimodal task and instruction data in Section 3.2. Finally, in Section 3.3, we illustrate the architecture of our SEEKER to support both long-context and short-context multimodal understanding.

### 3.1 USING IMAGE TOKENS TO ENCODE TEXT HELPS CONTEXT LENGTH EXTRAPOLATION

We follow the approach outlined in Xiong et al. (2023) to evaluate model's extrapolation capability in the First-Sentence-Retrieval task. In this task, models are required to retrieve the first sentence at a specific length. We conduct this synthetic task on various numbers of documents with different page counts. We probe the performance of GPT-4-Vision Image by feeding its images of documents and compare it with GPT-4-Vision Text and GPT-4, which receive extracted text using the OCR model Nougat Blecher et al. (2023). Nougat achieves over a 90 BLEU score on OCR text from scientific documents. All these models have a context length limit of $128k$ tokens.

On the left side of Figure 1, we visualize the Rouge-L Lin (2004) score in relation to the total number of pages of input documents, which range from 1 (approximately $1k$ text tokens) to 448

Table 1: **Long-Context Multimodal Task.** `Img/#In`: the number of input images, `Text Tok/#In` and `#Out`: the number of input and output text tokens. Full examples are presented in Appendix B.1.

| Task | Prompt Example | Img | Text Tok | |
|---|---|---|---|---|
| | | #In. | #In. | #Out. |
| *Long-Form Multi-Image Input* | | | | |
| Index | Which Image contains the given sentence? | 6.6 | 100.4 | 1.0 |
| SentRetrie | What is the first sentence on the first image? | 1.0 | 23.0 | 35.5 |
| ArxivQA | What is the main purpose of the article as stated in the abstract? | 8.2 | 13.9 | 35.0 |
| PassKey | What is the <PASSKEY> in the provided images? | 4.0 | 95.4 | 2.6 |
| *Long-Form Text Output* | | | | |
| ArxivVerb | Read the text in the image verbatim. | 1.0 | 10.0 | 1301.6 |
| WikiVerb | Read the text in the image verbatim. | 1.0 | 16.0 | 1107.1 |

(approximately $500k$ text tokens). We observe a significant performance degradation in models fed with text input. In contrast, without any additional changes, we see improved extrapolation when representing length text content with visual tokens by feeding images of documents directly to the model.

## 3.2 LONG-CONTEXT MULTIMODAL TASK

We mainly consider two categories of long-context multimodal capabilities, as outlined in Table 1: 1) Long-form multimodal input: This involves multiple text-rich images interleaved with text as the input context. 2) Long-form text output: This requires generating long text.

**Instruction Data for Long-Form Multi-Image Input**  First, we combine an arbitrary number of single-image visual instruction data Liu et al. (2023c) sourced from CC3M into the multi-image format for the intra-image reasoning task. This helps initiate model's capability of understanding sequences of images (e.g., $<img_1>$ *This image depicts a...* $<img_2>$ *This image shows a...*). We then curate inter-image reasoning instruction data from NLVR2 Suhr et al. (2019) (e.g., $<img_1>$ $<img_2>$ *Considering the images on both sides, is 'At least one of the televisions is turned off.' valid? Answer yes or no.*), Mimic CGD (e.g., $<img_1>$ $<img_2>$ *What's the difference between the two sinks in the images?*), and annotate multi-image conversation data on COCO images Lin et al. (2015) using GPT-4V (e.g., $<img_1>$ $<img_2>$ $<img_3>$ *How many birds are in all the provided images?*). To enable understanding of long-form text-rich image sequences, we collect compiled PDFs from arXiv documents. Each page from these documents is processed as images, ranging from 4 to 24 pages. We use GPT-4V to generate descriptive or conversational instruction data for these scientific documents. To further improve the model's understanding of each provided image, we create a multi-image text grounding task, requiring the model to ground the question to the referred image (e.g., $<img_1>$ $<img_2>$ ... $<img_8>$ *Which image contains the answer to the question / Which image contains the sentence...*).

**Instruction Data for Long-Form Text Output**  To enhance long-form text generation capabilities related to the given image, we propose a task that involves reading the text in the image verbatim (e.g., $<img_1>$ *Quote the text in the image verbatim.*). This challenging task requires the vision backbone to encode character-level image details and the language backbone to attend to the image token while producing very long text without hallucinating on previously generated content.

## 3.3 LONG-CONTEXT MULTIMODAL LARGE LANGUAGE MODEL

To enable long-context multimodal reasoning, our model architecture should: 1) encode multiple images interleaved with text, 2) align images and text at a fine-grained level, and 3) decode long texts that attend to extended multimodal contexts. The following paragraphs illustrate the design of our proposed SEEKER for this purpose.

**Long-Context Multi-Image Encoding**  For effective feature integration in scenarios involving multiple images, it is crucial to include image separators to concatenate text and image sequences as:

$$\text{Query} = \text{Query}_{\text{system}} + \sum_{i=1}^{N}\left(\mathbf{Q}_{\text{img},i} + \mathbf{Q}_{\text{txt},i}\right) \tag{1}$$

$$\mathbf{Q}_{\text{img},i} = \text{start}(\text{img}, i) + \text{content}(\text{img}, i) + \text{end}(\text{img}, i)$$

Specifically, we use start(img,i) and end(img,i) as special tokens '<|startofimgil|>' and '<|endofimgil|>' to distinguish the start and end of each image, respectively. We observe this strategy is essential for maintaining model performance, especially when training is limited to a small dataset of long-context multimodal instructions. The encoding process and the concatenation of the feature vectors of the input sequence can be described as:

$$t_i = \text{Enc}_{\text{t}}(\text{T}_{\text{i}}), v_i = \text{MLP}_{\text{v}\to\text{t}}(\text{Enc}_{\text{v}}(\text{I}_{\text{i}}))$$
$$Q = [t_0; v_1; t1; v_2; t2; \ldots; v_n; tn] \tag{2}$$

Here, $Enc_v$ encodes each image $i$ into a feature vector and projects it to the word embedding space. The concatenated vector $Q$ integrates sequences of image and text feature vectors, where $[;]$ denotes concatenation along the feature dimension.

Additionally, to preserve the model's capability with single-image data without necessitating re-finetuning, we introduce image-specific identifiers only during multi-image training and inference, while retaining the original prompt template for single-image contexts. Furthermore, incorporating image-index-aware question-answering instruction data enhances the model's ability to anchor its reasoning to specific images, enabling robust multi-image understanding and reasoning.

**Dense Image-Text Alignment**  We inherit the general image-text alignment from the pre-training image-text pairs. To enhance the visual representation of dense text in images, and improve the alignment between image and text representation of rendered text, we curate a visual-embedded task that renders text into visual space. Specifically, we render text paragraphs from Wikipedia into $1024 \times 1024$ images using Arial font, with sizes ranging from 18 to 30, providing various word densities per image. We observe that it is essential to start by learning image-text alignment at a sparse level (large font size, low word density) and gradually incorporate dense text-rendered image data. Task types we consider include question answering on multiple images rendered with text from Wikipedia, and reading the text verbatim from rendered images.

**Supervised Fine-tuning Strategy**  We aim to leverage sequential data processing to fine-tune models on a combination of textual and visual inputs, enabling them to generate coherent and contextually relevant responses based on both text and image data. In the domain of multimodal large language models, the autoregressive training objective is a pivotal technique, which can be formulated as follows:

$$p(X_o|Q) = \prod_{i=1}^{L} p_\theta(x_i|Q) \tag{3}$$

$$\mathcal{L}(\theta) = -\sum_{t=1}^{L} \log P(x_i|x_{<i}, Q; \theta)$$

where $x_i$ represents tokens with length $L$, $X_O$ denotes the target output given the features of multimodal queries $Q$, and $\theta$ denotes the model parameters. This loss function encourages the model to predict the next token in the sequence, given the previous visual and textual tokens.

## 4 IMPLEMENTATION DETAILS

### 4.1 MODEL ARCHITECTURE

The language model backbone of SEEKER is the DeepSeek LLM DeepSeek-AI et al. (2024), which has a design similar to LLaMA. It is supervised-finetuned on 2T tokens with additional DPO and surpasses LLaMA-2 and GPT-3.5 on numerous open-eval tasks. To enable to process high-resolution

images and ensure adept performance in real-world scenarios, we instruction-tune the stage-3 model from the DeepSeek-VL series of model DeepSeek-AI et al. (2024). The vision encoder of SEEKER-TINY is SigLIP, and the vision encoder of SEEKER is a hybrid of SigLIP-L Zhai et al. (2023) and SAM-B Kirillov et al. (2023). This enables processing $1024 \times 1024$ images into a fixed token length of $576$. This fixed token length for high-resolution image processing provides an optimal balance of fine-grained and compact visual representation. The adaptor used is a hybrid MLP, the same as in DeepSeek-VL Lu et al. (2024).

## 4.2 TRAINING

We use the AdamW Loshchilov & Hutter (2019) optimizer to train our models for 1 epoch with a batch size of 32. The learning rate is linearly warmed up during the first $5\%$ of steps to $1e-4$ and then reduced to zero using a cosine learning rate scheduler. The context sequence length is set to 4096 during instruction-tuning on single-image data. Both the vision-and-language pre-training data (e.g., MMC4 Zhu et al. (2023)) and single-image instruction-tuning data (e.g., ShareGPT4V Chen et al. (2023)) are adopted from DeepSeek-VL Lu et al. (2024). For continual training on our proposed long-context multimodal instruction data (Section 3.2), we set the maximum length to 8192 to accommodate a long sequence of images and long-form text output. We set the rank to 8 for low-rank adaptation (LoRA Hu et al. (2021)). Our SEEKER and SEEKER-TINY are trained on a single 8-A100-40G node for 30 hours and 12 hours, respectively.

## 4.3 EVALUATION

We consider four long-form multi-image input tasks: 1) `Index`: the multiple-choice image indexing task, given a sequence of images and a question, the model selects the option with the index of the image that contains the answer, 2) `SentRetrie`: the sentence retrieval task, given a sequence of images of rendered text sampled from Wikipedia, the model is required to retrieve the first sentence from the first image, 3) `ArxivQA`: the question answering on arxiv documents, the model is required to answer the question according to visual image of arxiv documents. 4) `PassKey`: the passkey retrieval task slightly modified for multimodal model, given the sentence with a masked word, the model need to answer what is the masked word by reading the visually-situated text content from arxiv document. We consider two long-form text output tasks: 1) `ArxivVerb`: extract text from the image of arxiv documents verbatim, 2) `WikiVerb`: extract text from the image of rendered text from Wikipedia verbatim. Details of each long-context multimodal task are introduced in Table 1, with more details presented in Appendix B.1.

Each long-context multimodal task contains $80$ diversified samples. We use the accuracy metric for the multiple-choice task (`Index`) and the Rouge-L score for all other text generation tasks. For standard multimodal tasks, which require fewer than four image inputs and text answers that are less than $400$ tokens. We use the accuracy metric for multiple-choice NLVR2 Suhr et al. (2019) test-public split and the BLINK Fu et al. (2024) validation split. We validate models on the official evaluation metrics and test splits for general single-image multimodal benchmarks, MMB EN, MMB CN (MMC) and Circular Eval for MMB (CCBench) Liu et al. (2024b), SEED Li et al. (2023a), AI2D Kembhavi et al. (2016), LLaVAB Liu et al. (2023c), ChartQA Masry et al. (2022), TextVQA Singh et al. (2019)). We follow the inference configurations in VLMEvalKit Contributors (2023).

## 5 MAIN RESULTS

### 5.1 LONG IMAGE AND TEXT CONTEXT

**Long-Form Multi-Image Input** In Table 2, SEEKER significantly surpass larger open-source MLLMs across all four long-form multi-image input tasks. We concatenate the images for models that can not handle image sequences. Additionally, SEEKER-TINY ranks second best. On average, our models also outperform the proprietary GPT-4V model. This indicates our auxiliary tasks, as detailed in Section 3.2, enhance the models' reasoning across multiple images and grounding content to specific images. Thus our models excel at handling long-context tasks involving long-form multiple text-rich image inputs.

Table 2: **Long Image and Text Context**.  ▨: proprietary models, ▨: the proposed models, `#Tok/Img`: the number of tokens per image. We report accuracy on multiple-choice task `Index`, and Rouge-L score for other tasks.

| Models | Params | #Tok/Img | Long-Form Multi-Image Input | | | | | Long-Form Text Output | | |
|---|---|---|---|---|---|---|---|---|---|---|
| | | | Index | SentR | ArxivQ | PassK | Avg | ArxivV | WikiV | Avg |
| **Close-source MLLMs** | | | | | | | | | | |
| GPT-4V OpenAI (2023b) | – | 85 | 32.50 | 71.10 | 45.19 | 27.16 | 43.98 | 32.58 | 5.96 | 19.27 |
| **Open-source MLLMs** | | | | | | | | | | |
| Qwen-VL-Chat Bai et al. (2023b) | 7B | – | 2.49 | 25.05 | 8.24 | 0.00 | 8.94 | 4.90 | 5.41 | 5.15 |
| LLaVA-1.5 Liu et al. (2023b) | 7B | 576 | 23.74 | 30.61 | 35.60 | 0.00 | 22.48 | 4.14 | 3.80 | 3.97 |
| LLaVA-Next Liu et al. (2024a) | 7B | 2880 | 17.49 | 34.35 | 20.50 | 0.00 | 18.08 | 22.33 | 22.94 | 22.63 |
| LLaVA-Next (Mistral) Liu et al. (2024a) | 7B | 2880 | 17.49 | 34.45 | 21.39 | 0.00 | 18.33 | 20.11 | 20.92 | 20.51 |
| DeepSeek-VL Lu et al. (2024) | 7B | 576 | 13.74 | 10.37 | 19.83 | 0.17 | 11.02 | 31.59 | 16.48 | 24.03 |
| IDEFICS2 Laurençon et al. (2024) | 8B | 64 | 10.83 | 63.46 | 9.68 | 0.13 | 21.02 | 12.12 | 5.93 | 9.02 |
| Monkey-Chat Li et al. (2023b) | 10B | – | 16.24 | 23.65 | 17.90 | 0.00 | 14.44 | 5.82 | 2.08 | 3.95 |
| LLaVA-1.5 Liu et al. (2023a) | 13B | 576 | 22.49 | 41.02 | 32.31 | 0.00 | 23.95 | 9.57 | 7.12 | 8.34 |
| LLaVA-Next Liu et al. (2024a) | 13B | 2880 | 11.24 | 37.55 | 15.60 | 0.00 | 16.09 | 27.14 | 31.05 | 29.09 |
| **Open-source Tiny MLLMs** | | | | | | | | | | |
| DeepSeek-VL Lu et al. (2024) | 1.3B | 576 | 14.99 | 10.46 | 21.29 | 0.15 | 11.72 | 20.06 | 10.43 | 15.24 |
| MiniCPM-V Hu et al. (2024) | 3B | – | 8.74 | 12.01 | 31.42 | 0.00 | 13.04 | 1.50 | 2.98 | 2.24 |
| **Ours** | | | | | | | | | | |
| SEEKER-TINY | 1.3B | 576 | **33.74** | 66.99 | **42.68** | 24.99 | 42.10 | 23.52 | 25.33 | 24.42 |
| SEEKER | 7B | 576 | 27.49 | **71.33** | 42.35 | **37.91** | **44.77** | 31.85 | **34.98** | **33.41** |

**Long-Form Text Output**  In Table 2, our SEEKER achieves the best performance for long-context tasks requiring long-form text output. On average, LLaVA-Next Liu et al. (2024a)-13B also performs well, likely because these tasks usually require a single image. Its feature of splitting images into four tiles as additional 2304 image tokens, combined with the original image, greatly enhances its ability to capture visual details. This is particularly beneficial for verbatim tasks involving Arxiv and Wikipedia content rendered in the image. Meanwhile, DeepSeek-VL Lu et al. (2024) achieves the best scores among other open-source 7B MLLMs , primarily due to its alignment of image and text by enforcing text reading from a large scale of visual-situated real-world data, such as documents and PDFs. By incorporating our small-scale verbatim task data, which includes images rendered with text of various font sizes, into the instruction-tuning stage, our models achieve a 38.1% performance improvement.

**Fix-length Image Tokens are more Expressive than Text Tokens**  If a model can interpret text within images, it confirms that this method is a valid way to present information. Additionally, if the model requires fewer image tokens than text tokens to understand the text, this indicates that pixels can represent text more compactly. To investigate this, we conduct a probing task involving question-answering using various pages of documents fed into the model, as shown in Table 3. Notably, in this task, we use a version of our SEEKER with the same context length as the compared model, which

Table 3: Probing Question Answering with Varying Page Context: Our SEEKER model seeks more accurate text answers within compact image tokens of image sequences compared to OCR-based approaches with the same context length. $p$ stands for the range of page numbers of the document.

| Models | Input Type | ArxivQA | | | | |
|---|---|---|---|---|---|---|
| | | p=4:6 | p=6:8 | p=8:10 | p=10:12 | Avg |
| *LLM* | | | | | | |
| DeepSeek-LLM | OCR Txt | 35.79 | 35.74 | 36.00 | 29.99 | 34.38 |
| SEEKER -LLM | OCR Txt | **45.26** | 46.17 | 50.57 | 39.18 | 45.29 |
| *MLLM* | | | | | | |
| DeepSeek-VL | Seq Img | 29.30 | 37.97 | 36.67 | 28.38 | 33.08 |
| SEEKER | Seq Img+OCR Txt | 35.30 | 41.22 | 40.73 | 33.49 | 37.68 |
| SEEKER | Seq Img | 44.43 | **50.81** | **58.10** | **39.95** | **48.32** |

is 4,096 tokens. Our observations indicate that when the text token count is up to around 4,000, the response accuracy remains within the context length limit of 4,096 tokens without performance degradation for the language model (LLM). When the text token count exceeds 4,000 but the image token count remains below 4,000, the vision-language model (VLM) outperforms the LLM by 4 to 8 percentage points. However, when the image token count exceeds 4,000, the performance of the VLM also declines, though it remains slightly superior to that of the LLM.

Table 4: **Short Image and Text Context**. ▢ : proprietary models, ▢ : the proposed models. We compare our SEEKER with other MLLMs on multi-image and single-image benchmarks.

| Models | Multi-Image | | | Single-Image | | | | | | | | |
|---|---|---|---|---|---|---|---|---|---|---|---|---|
| | NLVR2 | BLINK | Avg | MMB | MMC | SEED | CCBench | AI2D | LLaVAB | ChartQA | TextVQA | Avg |
| **Close-source MLLMs** | | | | | | | | | | | | |
| GPT-4V OpenAI (2023b) | 71.7 | 51.1 | 61.4 | 75.1 | 74.4 | 71.6 | 46.5 | 75.9 | 93.1 | 78.5 | 78.0 | 60.3 |
| **Open-source MLLMs** | | | | | | | | | | | | |
| Qwen-VL-Chat Bai et al. (2023b) | 30.8 | 28.1 | 29.5 | 60.6 | 56.3 | 64.8 | 41.2 | 63.0 | 67.7 | 49.8 | 60.7 | 58.0 |
| LLaVA-1.5-7B Liu et al. (2023a) | 61.7 | 37.1 | 49.4 | 65.2 | 59.0 | 65.8 | 27.5 | 55.5 | 61.8 | 17.8 | 45.4 | 49.8 |
| LLaVA-Next-7B Liu et al. (2024a) | 58.7 | 41.2 | 49.9 | 67.4 | 62.3 | 69.6 | 24.3 | 67.0 | 72.7 | 55.4 | 64.4 | 60.4 |
| LLaVA-Next-7B (Mistral) Liu et al. (2024a) | 43.5 | 37.5 | 40.5 | 69.5 | 61.3 | **72.4** | 30.0 | _69.0_ | 67.8 | 51.8 | 65.2 | 63.1 |
| DeepSeek-VL-7B Lu et al. (2024) | 46.6 | 40.9 | 43.7 | **74.1** | 71.4 | 70.4 | _51.7_ | 65.3 | 77.8 | 59.1 | 64.9 | _66.8_ |
| IDEFICS2-8B Laurençon et al. (2024) | **79.9** | **46.8** | **63.4** | 75.3 | 67.3 | 71.9 | 37.6 | 72.3 | 49.1 | 24.36 | 68.9 | 66.3 |
| Monkey-Chat-10B Li et al. (2023b) | 66.0 | 40.5 | 53.3 | 71.0 | 65.8 | 68.9 | 48.4 | 68.5 | 60.5 | _59.5_ | _65.5_ | 63.5 |
| LLaVA-1.5-13B Liu et al. (2023a) | 66.2 | _42.7_ | 54.4 | 69.2 | 65.0 | 68.2 | 30.4 | 61.1 | 66.1 | 18.2 | 48.9 | 53.4 |
| LLaVA-Next-13B Liu et al. (2024a) | 64.3 | 42.6 | 53.4 | 70.7 | **79.0** | _71.9_ | 28.8 | **72.2** | 73.9 | **61.4** | **66.9** | 65.6 |
| **Open-source Tiny MLLMs** | | | | | | | | | | | | |
| DeepSeek-VL-1.3B Lu et al. (2024) | 61.3 | 38.8 | 50.1 | 64.0 | 62.9 | 66.0 | 37.6 | 51.5 | 51.1 | 47.4 | 57.8 | 54.8 |
| MiniCPM-V-3B Hu et al. (2024) | 63.1 | 40.0 | 51.5 | 67.9 | 62.6 | 65.6 | 41.4 | 56.3 | 51.3 | 44.2 | 56.6 | 55.7 |
| **Ours** | | | | | | | | | | | | |
| SEEKER-TINY -1.3B | 69.9 | 40.5 | 55.2 | 64.8 | 63.7 | 66.0 | 37.3 | 49.0 | **81.7** | 45.4 | 56.3 | 58.0 |
| SEEKER -7B | _72.4_ | 42.1 | _57.2_ | _74.0_ | _72.6_ | 71.1 | **52.0** | 64.6 | _79.3_ | 58.3 | 65.3 | **67.1** |

## 5.2 GENERAL MULTIMODAL UNDERSTANDING BENCHMARK

We aim to evaluate the general multimodal understanding and reasoning capabilities of our model in comparison with state-of-the-art models in the field. In Table 4, our model, SEEKER , demonstrates performance on par with other models of similar size when tested on short-context multi-image tasks. This consistency in performance is noteworthy, given that our model excels in these tasks without requiring significant additional resources or tuning.

Moreover, even though we did not explicitly include general single-image instruction data during the continual instruction tuning phase for long-context tasks, our model still retains competitive performance. In fact, SEEKER performs on par with other MLLMs in this domain and even outperforms all other models on certain tasks. This ability to maintain performance, despite the absence of further instruction tuning data, can be attributed to our approach of employing a distinct image identifier for multi-image processing, while continuing to use the single-image template during inference. This strategy allows the model to handle multi-image tasks efficiently without compromising its performance on single-image tasks.

## 6 ANALYSIS

### 6.1 CONTEXT LENGTH EXTRAPOLATION

We analyze the effectiveness of using image tokens versus OCR text tokens for image representation. The density plot in Figure 4 illustrates the distribution of token counts for both methods. The Image token representation is notably more compact, with a significant peak at lower token counts, whereas the OCR-text displays a broader distribution with higher counts. This variation shows that OCR-text length can be vulnerable and uncontrollable in images rich in text, often leading to wide-ranging token counts. In contrast, image tokens maintain a consistent token length regardless of textual density. With a model context length set to 8192 tokens, image tokens are handled 100% of the time without truncation, whereas OCR-text frequently exceeds this limit, achieving only 66.25% execution success without truncation. Meanwhile, truncating OCR text compromises performance

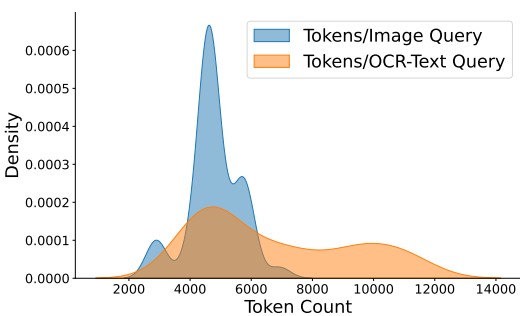

Figure 4: Density plot comparing token counts for image token (blue) and OCR-text (orange) representations. Image tokens are more compact than text, fitting well within 8192 context length.

as shown in Table 3. This highlights the advantages of image tokens for predictable and efficient encoding of long multimodal contexts.

## 6.2 INFERENCE EFFICIENCY

In addition to its context length extrapolation capability, our model SEEKER solves long-context multimodal tasks more efficiently compared to the OCR-based approach. For example, when comparing the inference time cost of SEEKER with and without OCR, the latter first extracts long text from multiple images and then feeds text into SEEKER. By eliminating the time-consuming OCR step, our model achieves a significant reduction in inference time. Specifically, in the longest context scenario, SEEKER is approximately three times faster than OCR-based approach, showcasing the substantial time efficiency.

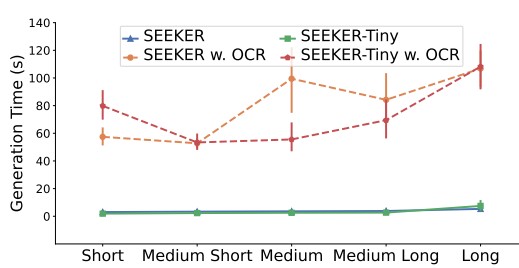

Figure 5: Generation times for SEEKER and SEEKER-TINY with and without OCR.

## 6.3 TRADEOFF OF COMPACT CONTEXT LENGTH AND HIGH RESOLUTION

In Figure 6, we show GPT-4-Vision with low and high resolution setting on first-sentence-retrieval. With high-resolution mode, more tokens will be used to represent the same image. Although high-resolution usually brings more details and better performance, we can see it tradeoffs capability of extrapolating long page document understanding. And thus only GPT-4-Vision low-resolution model preserves the performance in this probing task. On the right we can see that high-resolution usually take more image tokens to represent text-rich image than text tokens of OCR-extracted content, and thus even drops more quickly than feeding text.

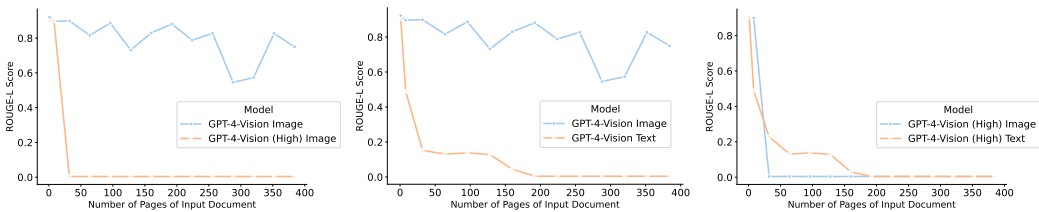

Figure 6: Performance plot on First-Sentence-Retrieval task. GPT-4-Vision Image and GPT-4-Vision (High) Image directly process the long-context information in image, the *High* refers to high resolution mode compared with low one. GPT-4-Vision Text represents the approach to process long-context information in OCR-extracted content.

## 6.4 QUALITATIVE SHOWCASES

Figure 7 showcases the SEEKER model's performance on three tasks, emphasizing its long-context capabilities. In the verbatim generation task, SEEKER read text from the arXiv paper, indicating its coherent narratives given extended multimodal context. For the first sentence retrieval task, it efficiently navigated and extracted key sentences from extensive texts without utilizing the OCR model. In the task of reasoning across multiple images, the model effectively grounds the text in the specific image as required. At the bottom of Figure 7, we observe that SEEKER can also generalize to multi-frame video understanding. We compare SEEKER-7B with DeepSeek-VL-7B on identifying the document titles in Table 5. SEEKER excels at capturing character-level details. These results illustrate SEEKER's proficiency in handling long-context multimodal tasks, marking a significant advancement in MLLMs.

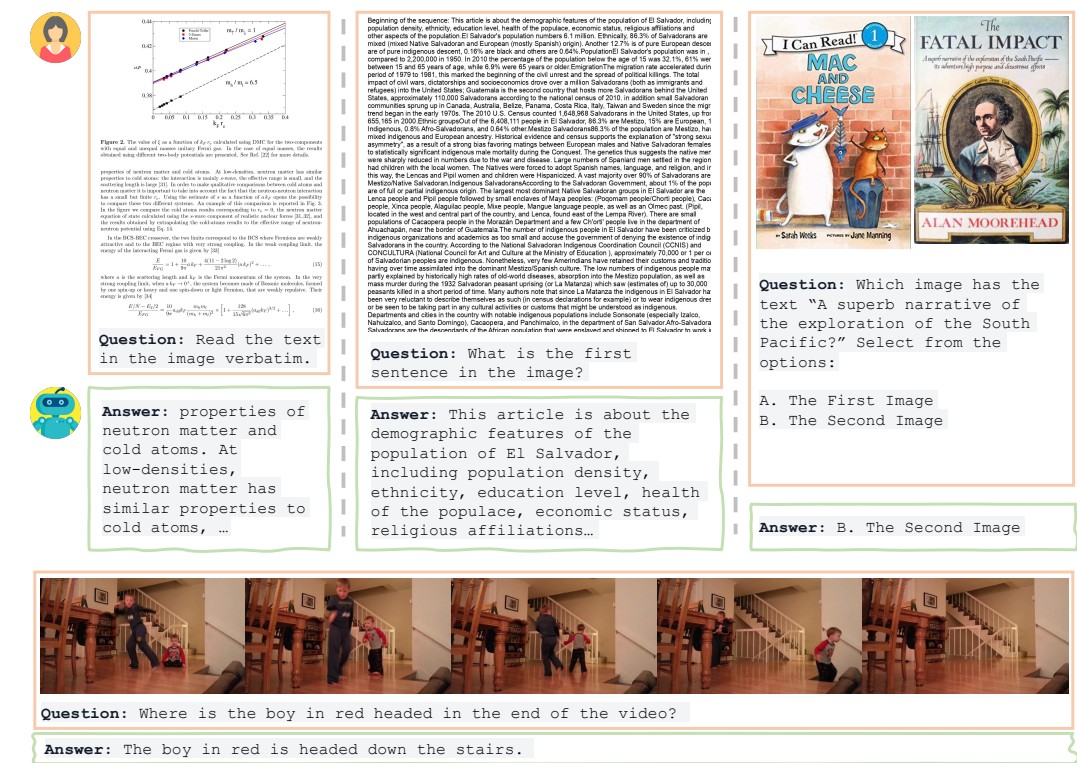

Figure 7: Showcases of the SEEKER 's performance on verbatim text generation, sentence retrieval, multi-image reasoning, and video question answering, demonstrating its long-context understanding.

Table 5: Comparisons of MLLMs' Instruction-Following Character-Level Recognition.

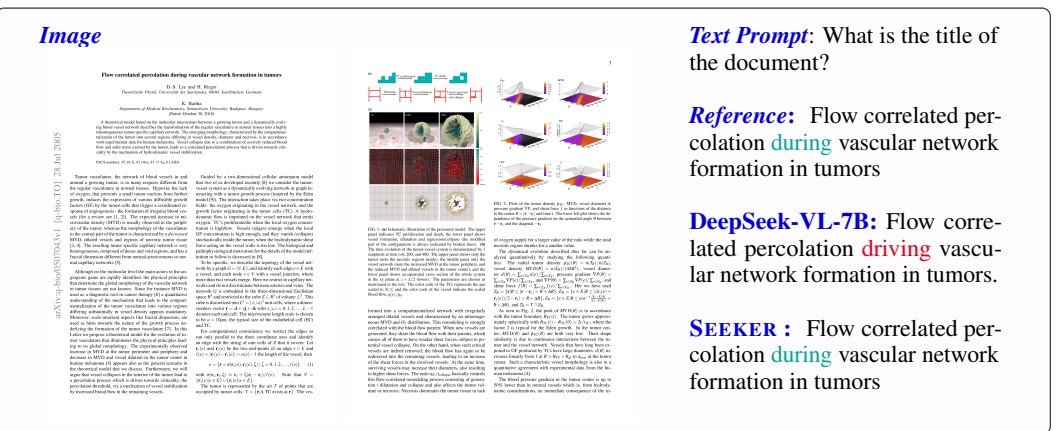

# 7 CONCLUSION

In this paper, we present SEEKER , which advances the field of long-context comprehension in multimodal large language models. By enhancing the processing of lengthy texts presented in visual formats and continual instruction-tuning on extended context tasks, SEEKER surpasses existing multimodal large language models in handling extensive multimodal contexts. Our SEEKER also shows efficiency compared with OCR-based approach in terms of better long context extrapolation and inference efficiency.

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

# Part I

# Appendix

## Table of Contents

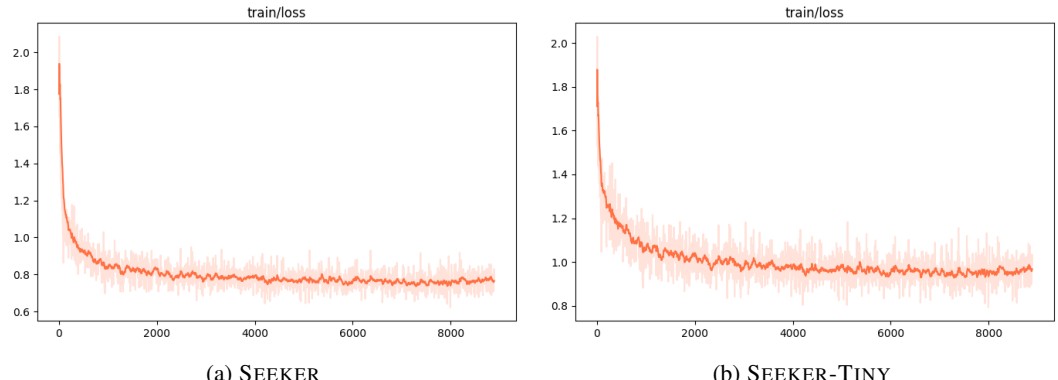

(a) SEEKER                                       (b) SEEKER-TINY

Figure 8: Training Loss Curve.

## A   IMPLEMENTATION DETAILS OF SEEKER

### A.1   TRAINING LOSS CURVE

In Figure 8, we show the training loss curve of our SEEKER and SEEKER-TINY . Though both model have a quick loss drop initially, we observe a smoother and more consistent decrease of SEEKER than SEEKER-TINY . In the end, SEEKER stabilizes at a lower loss value, suggesting its potentially better generalization capabilities than SEEKER-TINY .

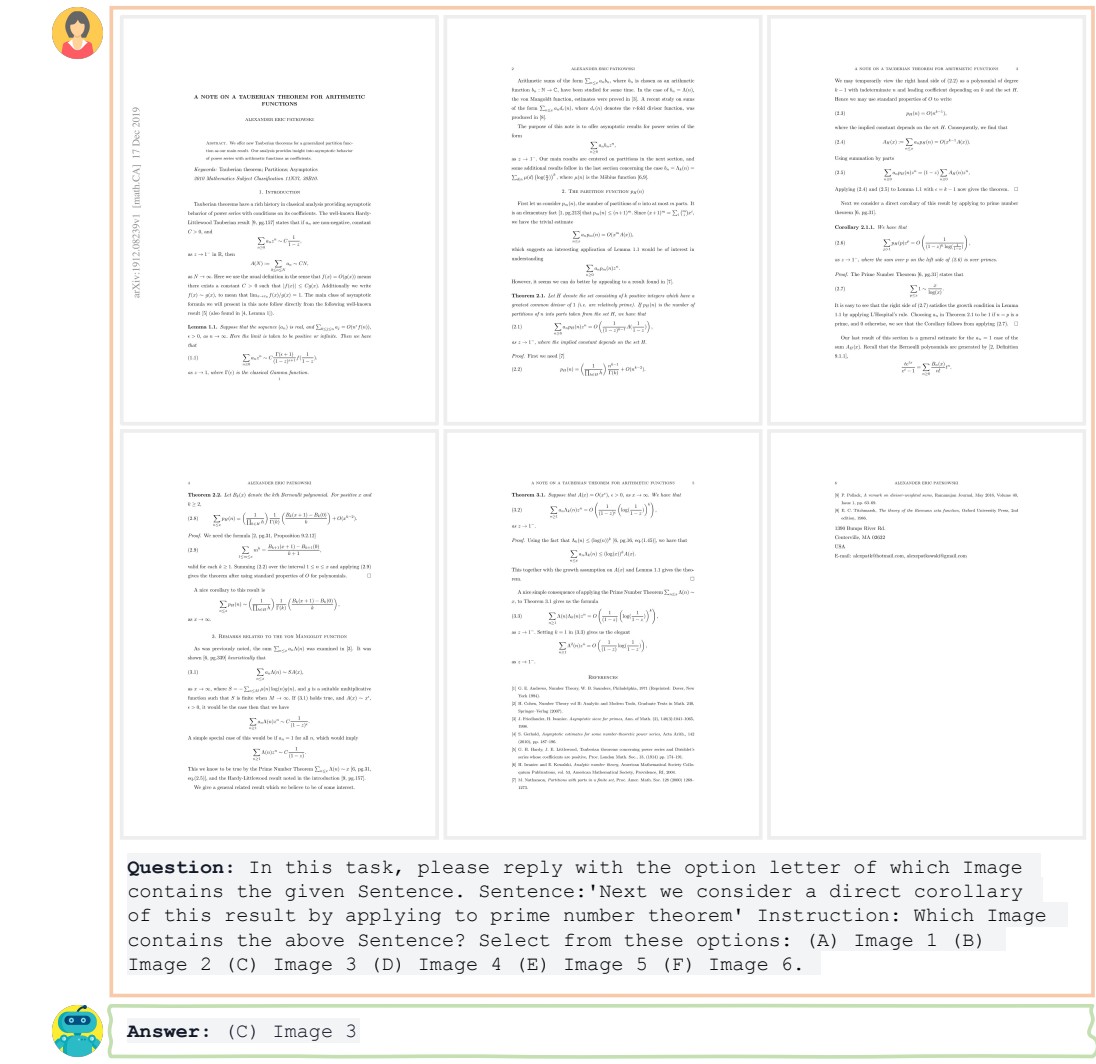

**Question:** In this task, please reply with the option letter of which Image contains the given Sentence. Sentence:'Next we consider a direct corollary of this result by applying to prime number theorem' Instruction: Which Image contains the above Sentence? Select from these options: (A) Image 1 (B) Image 2 (C) Image 3 (D) Image 4 (E) Image 5 (F) Image 6.

**Answer:** (C) Image 3

Figure 9: Task `Index`.

# B LONG-CONTEXT MULTIMODAL TASKS

## B.1 TASK EXAMPLES

In Section 3.2, we first introduce multimodal long-context tasks categorized in long-form multi-image input and long-form text output. And in Figure 9-14, we visualize full task examples.

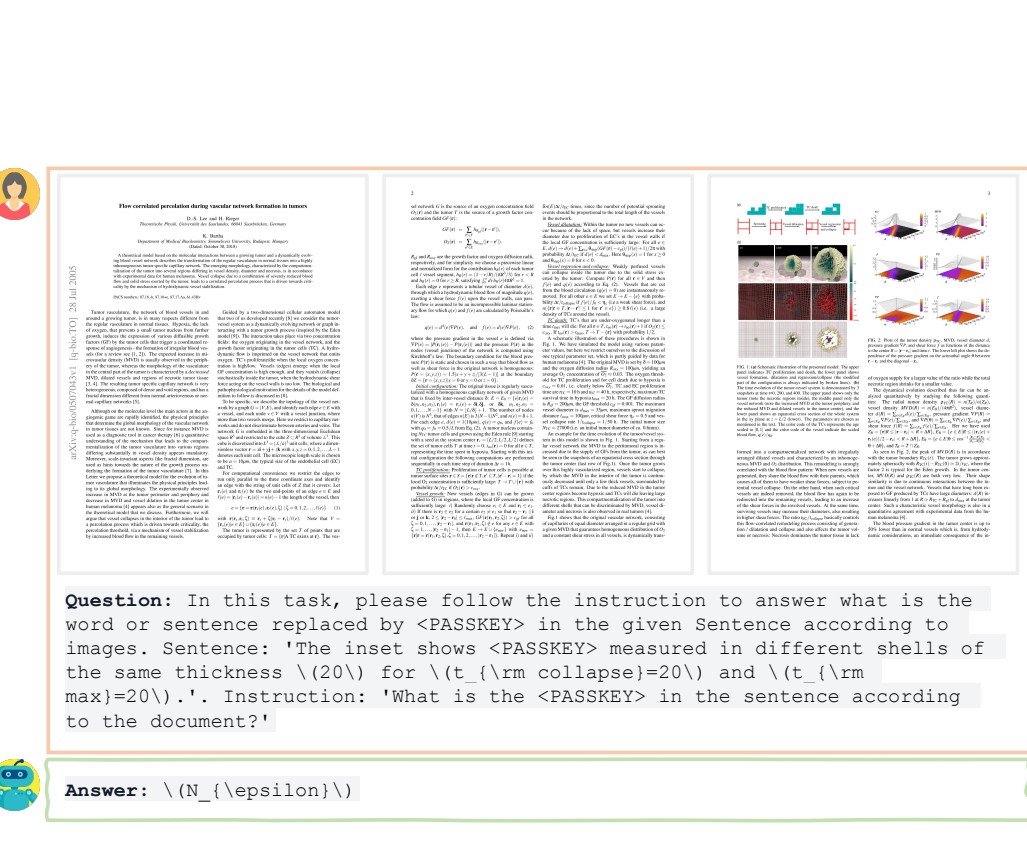

**Question:** In this task, please follow the instruction to answer what is the word or sentence replaced by <PASSKEY> in the given Sentence according to images. Sentence: 'The inset shows <PASSKEY> measured in different shells of the same thickness \(20\) for \(t_{\rm collapse}=20\) and \(t_{\rm max}=20\).'. Instruction: 'What is the <PASSKEY> in the sentence according to the document?'

**Answer:** \(N_{\epsilon}\)

Figure 10: Task `PassKey`.

**Question:** What is the definition of a Latin square according to Definition 1.1?

**Answer:** According to Definition 1.1, a Latin square of order n is an n × n matrix where each row and each column is a permutation of elements of [n].

Figure 11: Task `ArxivQA`.

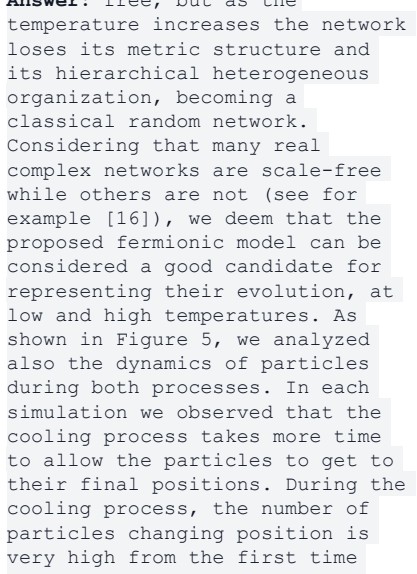

Figure 12: Task `ArxivVerbatim`.

Beginning of the sequence: Aristotle (; Aristotélēs, ; 384–322 BC) was a Greek philosopher and polymath during the Classical period in Ancient Greece. Taught by Plato, he was the founder of the Lyceum, the Peripatetic school of philosophy, and the Aristotelian tradition. His writings cover many subjects including physics, biology, zoology, metaphysics, logic, ethics, aesthetics, poetry, theatre, music, rhetoric, psychology, linguistics, economics, politics, meteorology, geology and government. Aristotle provided a complex synthesis of the various philosophies existing prior to him. It was above all from his teachings that the West inherited its intellectual lexicon, as well as problems and methods of inquiry. As a result, his philosophy has exerted a unique influence on almost every form of knowledge in the West and it continues to be a subject of contemporary philosophical discussion.Little is known about his life. Aristotle was born in the city of Stagira in Northern Greece. His father, Nicomachus, died when Aristotle was a child, and he was brought up by a guardian. At seventeen or eighteen years of age he joined Plato's Academy in Athens and remained there until the age of thirty-seven (c. 347 BC). Shortly after Plato died, Aristotle left Athens and, at the request of Philip II of Macedon, tutored Alexander the Great beginning in 343 BC. He established a library in the Lyceum which helped him to produce many of his hundreds of books on papyrus scrolls. Though Aristotle wrote many elegant treatises and dialogues for publication, only around a third of his original output has survived, none of it intended for publication.Aristotle's views profoundly shaped medieval scholarship. The influence of physical science extended from Late Antiquity and the Early Middle Ages into the Renaissance, and were not replaced systematically until the Enlightenment and theories such as classical mechanics were developed. Some of Aristotle's zoological observations found in his biology, such as on the hectocotyl (reproductive) arm of the octopus, were disbelieved until the 19th century. He also influenced Judeo-Islamic philosophies (800–1400) during the Middle Ages, as well as Christian theology, especially the Neoplatonism of the Early Church and the scholastic tradition of the Catholic Church. Aristotle was revered among medieval Muslim scholars as "The First Teacher", and among medieval Christians like Thomas Aquinas as simply "The Philosopher", while the poet Dante called him "the master of those who know". His works contain the earliest known formal study of logic, and were studied by medieval scholars such as Peter Abelard and John Buridan.Aristotle's influence on logic continued well into the 19th century. In addition, his ethics, though always influential, gained renewed interest with the modern advent of virtue ethics.Aristotle has been called "the father of logic", "the father of biology", "the father of political science", "the father of zoology", "the father of embryology", "the father of natural law", "the father of scientific method", "the father of rhetoric", "the father of psychology", "the father of realism", "the father of criticism", "the father of individualism", "the father of teleology", and "the father of meteorology".LifeIn general, the details of Aristotle's life are not well-established. The biographies written in ancient times are often speculative and historians only agree on a few salient points.Aristotle, whose name means "the best purpose" in Ancient Greek, was born in 384 BC in Stagira, Chalcidice, about 55 km (34 miles) east of modern-day Thessaloniki. His father, Nicomachus, was the personal physician to King Amyntas of Macedon. While he was young, Aristotle learned about biology and medical information, which was taught by his father. Both of Aristotle's parents died when he was about thirteen, and Proxenus of Atarneus became his guardian. Although little information about Aristotle's childhood

**Question:** Read the text in the image verbatim.

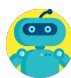

**Answer:** Aristotle (; Aristotélēs, ; 384–322 BC) was a Greek philosopher and polymath during the Classical period in Ancient Greece. Taught by Plato, he was the founder of the Lyceum, the Peripatetic school of philosophy, and the Aristotelian tradition. His writings cover many subjects including physics, biology, zoology, metaphysics, logic, ethics, aesthetics, poetry, theatre, music, rhetoric, psychology, linguistics, economics, politics, meteorology, geology and government. Aristotle provided a complex synthesis of the various philosophies existing prior to him.

Figure 13: Task `WikiVerbatim`.

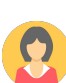

Beginning of the sequence: An American in Paris is a jazz-influenced orchestral piece by American composer George Gershwin first performed in 1928. It was inspired by the time that Gershwin had spent in Paris and evokes the sights and energy of the French capital during the Années folles.Gershwin scored the piece for the standard instruments of the symphony orchestra plus celesta, saxophones, and automobile horns. He brought back four Parisian taxi horns for the New York premiere of the composition, which took place on December 13, 1928, in Carnegie Hall, with Walter Damrosch conductin the New York Philharmonic. It was Damrosch who had commissioned Gershwin to write his Concerto in F following the earlier success of Rhapsody in Blue (1924). He completed the orchestration on November 18, less than four weeks before the work's premiere. He collaborated on the original program notes with critic and composer Deems Taylor.BackgroundAlthough the story is likely apocryphal, Gershwin is said to have been attracted by Maurice Ravel's unusual chords, and Gershwin went on his first trip to Paris in 1926 ready to study with Ravel. After his initial student audition with Ravel turned into a sharing of musical theories, Ravel said he could not teach him, saying, "Why be a second-rate Ravel when you can be a first-rate Gershwin?"Gershwin strongly encouraged Ravel to come to the United States for a tour. To this end, upon his return to New York, Gershwin joined the efforts of Ravel's friend Robert Schmitz, a pianist Ravel had met during the war, to urge Ravel to tour the U.S. Schmitz was the head of Pro Musica, promoting Franco-American musical relations, and was able to offer Ravel a $10,000 fee for the tour, an enticement Gershwin knew would be important to Ravel.Gershwin greeted Ravel in New York in March 1928 during a party held for Ravel's birthday by Éva Gauthier. Ravel's tour reignited Gershwin's desire to return to Paris, which he and his brother Ira did after meeting Ravel. Ravel's high praise of Gershwin in an introductory letter to Nadia Boulanger caused Gershwin to seriously consider taking much more time to study abroad in Paris. Yet after he played for her, she told him she could not teach him. Boulanger gave Gershwin basically the same advice she gave all her accomplished master students: "What could I give you that you haven't already got?" This did not set Gershwin back, as his real intent abroad was to complete a new work based on Paris and perhaps a second rhapsody for piano and orchestra to follow his Rhapsody in Blue. Paris at this time hosted many expatriate writers, among them Ezra Pound, W. B. Yeats, Ernest Hemingway, and artist Pablo Picasso.CompositionGershwin based An American in Paris on a melodic fragment called "Very Parisienne", written in 1926 on his first visit to Paris as a gift to his hosts, Robert and Mabel Schirmer. Gershwin called it "a rhapsodic ballet"; it is written freely and in a much more modern idiom than his prior works.Gershwin explained in Musical America, "My purpose here is to portray the impressions of an American visitor in Paris as he strolls about the city, listens to the various street noises, and absorbs the French atmosphere."The piece is structured into five sections, which culminate in a loose ABA format. Gershwin's first A episode introduces the two main "walking" themes in the "Allegretto grazioso" and develops a third theme in the "Subito con brio". The style of this A section is written in the typical French style of composers Claude Debussy and Les Six. This A section featured duple meter, singsong rhythms, and diatonic melodies

**Question:** What is the first sentence in the image?

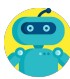

**Answer:** An American in Paris is a jazz-influenced orchestral piece by American composer George Gershwin first performed in 1928.

Figure 14: Task SentRetrie.

# C  DISCUSSION

## C.1  LIMITATIONS

While our model, SEEKER, has made significant strides in processing extended-context multimodal inputs, it encounters several critical limitations that require deeper investigation. The process of compressing textual information into visual tokens, although efficient, may inadvertently overlook precise textual understanding. Future endeavors should focus on developing hybrid encoding strategies that balance token compression with the preservation of essential information. Additionally, SEEKER could inadvertently learn and perpetuate biases present in its training data. It is imperative that further research is conducted to identify, understand, and address these biases, ensuring the model's equity and inclusiveness.

## C.2  SOCIETAL IMPACT

By integrating visual tokens with textual data, SEEKER addresses the limitations of traditional models and supports the handling of longer input sequences. This innovation could transform various sectors, improving information accessibility and retrieval systems across academic research, legal document analysis, and extensive data processing tasks. Particularly beneficial in educational and professional environments, SEEKER enables rapid and accurate extraction of vast informational content, fostering better decision-making and knowledge dissemination. However, this advancement might exacerbate information disparities if not equitably accessible. Steps should be taken to make sure it is both affordable and available to everyone.

