# OpenReview forum: "Compact Multimodal Context Represenations Using Visual Tokens"
_ICLR.cc/2025/Conference — Submitted to ICLR 2025_

### Official Review · Reviewer_qvLC · 2024-10-25

**Soundness:** 1
**Presentation:** 2
**Contribution:** 2
**Rating:** 3
**Confidence:** 5

**Summary:**

The authors present a method that improves long-context comprehension in vision large language models (VLLM) by converting long text input into visual text rendered on images, reducing the number of tokens. This approach reduces the computational cost of the self-attention operation.

**Strengths:**

1. Experimental results validate the method's effectiveness across selected tasks.
2. The writing is clear and straightforward, making it easy to follow.

**Weaknesses:**

1. The experiments focus on relatively simple tasks, which favor the proposed method. However, rendering text on images may result in information loss with small fonts or low resolution, limiting performance on more complex tasks that require precise context understanding or long text generation.
2. The criteria for ensuring that the image token sequence is shorter than the text token sequence are not discussed, which is crucial for real-world applications. Metrics like tokens per image or pixels per token might be useful for setting a clear threshold.
3. Text-only LLMs with larger context windows (e.g., 128k tokens) are not evaluated as a baseline, despite being a practical option for real-world applications.
4. Key details of the text rendering process and OCR text assembly are not provided. Important factors such as the average number of text tokens rendered per image and whether OCR layout bounding boxes are utilized are missing.

**Questions:**

Please refer to the weakness.

In addition. The assumptions regarding whether the input text is available or unavailable are unclear.

1. If the text is available, is the number of visual tokens less than true text tokens as well? (weakness 3).
2. If the text is unavailable, the experiments essentially compare VLLM with OCR+LLM, a scenario that has already been extensively studied. The advantage of the proposed method over existing VLLMs may be due to training on a specific rendering format rather than a true improvement in handling long contexts?

---

### Official Review · Reviewer_nbtq · 2024-11-04

**Soundness:** 2
**Presentation:** 3
**Contribution:** 2
**Rating:** 5
**Confidence:** 4

**Summary:**

This study focuses on improving how multimodal large language models (MLLMs) handle reasoning with multiple images. Although many (M)LLMs exist, they often struggle with multi-image reasoning tasks. To address this, the authors propose several tasks designed to enhance model performance, such as "intra-image reasoning", "inter-image reasoning", "long-form text-rich image understanding", and "multi-image text grounding". The authors curated a specialized training dataset and introduced a new model, Seeker, built on the DeepSeek-VL model. Through evaluations across six specific tasks, Seeker demonstrated superior performance over existing models, including on single-image benchmarks, ensuring it retained its effectiveness on existing well-established tasks.

**Strengths:**

1. The paper effectively highlights existing models' difficulties with text-based multi-image document reasoning. Figure 1, in particular, will likely be notable for readers, including the OpenAI research team. Additionally, this challenge is highly relevant from an industry perspective, as document-centered tasks remain central to robotic process automation, a field of ongoing interest.

2. The paper is well-written and organized, with a clear background section and relevant citations, making it accessible and easy to understand.

3. The tasks proposed in this study and the resulting dataset contribute to improving model performance. Regardless of any debate on the paper's claims about modality, this approach appears valuable for advancing various vision-language model studies. If the authors can share the data or code, it would greatly benefit the community; if this is not feasible, providing detailed data generation steps in the appendix is recommended.

**Weaknesses:**

1. In Table 2, the low performance of existing models on the "Long-Form Multi-Image Input" evaluation may indicate that these models lack generalization for the proposed tasks rather than it being a definitive advantage of a particular modality. Before concluding any superiority in modality, further validation seems necessary. Specifically, I would strongly recommend filling the "img" in Section 3.2 with OCRed text, then training the baseline model on this version and performing the evaluation in a text-based approach. This could clarify the generalization issue and offer a more robust analysis of the effectiveness of vision representation in this context.

2. It is challenging to assess how much the proposed task-specific datasets in Section 3.2 contribute to the final evaluation metrics (Section 4.3). To clarify this aspect, an ablation study based on the baseline model, DeepSeek-VL, is recommended.

**Questions:**

1. In Table 2, was Seeker trained on the dataset in Section 3.2, while the comparison models were evaluated in a zero-shot setting?

2. In Figure 3, should the "4k context length" be updated to "8k context length"?

3. Clarifying labels for (1) and (2) in Figure 3 would be helpful.

4. In Section 3.1, could you share the reason for testing with GPT-4(V)? Would similar trends appear with well-known open-source models or the baseline model, DeepSeek-VL?

5. In Table 1, it seems unusual that "SentR" performs better than "Index," as I expected "Index" to be the more straightforward task.

6. In Table 1, "SentR" appears to have been conducted as a single-image task. Please clarify why it wasn't set up with multiple images.

7. There doesn't seem to be an analysis of font type and size, which could affect visual text understanding. These factors notably impact OCR systems, and the model's performance may vary depending on the font choice and rendering.

8. Beyond the difference in corpora, is there a variation in difficulty between ArxivVerb and WikiVerb? In Table 2, some open-source models seem to struggle more with WikiVerb.

---

### Official Review · Reviewer_sLTS · 2024-11-05

**Soundness:** 2
**Presentation:** 2
**Contribution:** 2
**Rating:** 5
**Confidence:** 3

**Summary:**

The paper introduces SEEKER, a multimodal large language model optimized for efficiently handling long-form multimodal content. By leveraging visual tokens instead of traditional text tokens for text representation, SEEKER achieves compact encoding, enabling better long-context understanding and faster processing in tasks involving extensive image-text inputs and outputs, outperforming existing models.

**Strengths:**

1.  It introduces a unique tokenization method of using visual tokens to represent textual content, effectively compacting long-context data and allowing for efficient processing within a limited token length, which leads to faster inference times.

2. SEEKER outperforms existing multimodal models on multiple long-context tasks, as shown in comparative evaluations across various datasets, including tasks involving multiple images and long-form text outputs.

3. The paper rigorously tests SEEKER on a wide range of multimodal tasks, providing detailed comparisons with proprietary and open-source models, which validates the model’s generalizability and robustness across diverse scenarios.

**Weaknesses:**

1.  The writing is poor. The citation format is incorrect (Maybe it should be citep), the method section is confusing, and I don't know how to get a compact embedding for a long-text input. The mathematical formulas are not standardized.

2.   In Figure 1, why use the ROUGE-L score rather than other metrics?

3.   By converting text to visual tokens, there is a risk of losing fine-grained linguistic details that could affect accuracy in tasks requiring precise textual analysis, as visual tokens might not capture subtleties as accurately as text tokens.

4.   The paper does not explore interpretability mechanisms, such as visual attention maps or token significance, which could help in understanding how SEEKER handles image-text integration, especially in complex contexts.

**Questions:**

As shown in weaknesses.

---

### Meta-Review · Area_Chair_6GtZ · 2024-12-20

**Metareview:**

This paper focuses on improving multimodal large language models by encoding long textual inputs into visual pixel space to efficiently handle long-form multimodal content. However, it has several major issues, including poor writing quality, unclear descriptions of the methodology, missing critical technical details, non-standard mathematical formulations, and insufficient comparative experiments and evaluations. Notably, reviewers highlighted concerns such as converting long text into visual tokens, which may introduce subtle information loss and lead to suboptimal performance on tasks requiring precise textual analysis. Additionally, the paper lacks interpretability analyses on how the model integrates image and text information, does not compare against text models with larger context windows, and omits ablation studies to validate the method's effectiveness.

Given that these significant shortcomings remain unaddressed, and no feedback has been provided, this paper does not currently meet the standards for acceptance.

**Additional Comments On Reviewer Discussion:**

During the review process, several critical issues were raised by the reviewers regarding the submission. Reviewers highlighted concerns about the poor writing quality, incorrect citation formats, and non-standard mathematical formulations, which hindered the clarity and understanding of the methodology. Reviewer qvLC pointed out that the experiments were conducted on simple tasks that favored the proposed method, and there was a lack of discussion on ensuring that the image token sequence is shorter than the text token sequence, which is crucial for practical applications. Reviewer nbtq questioned the generalization capability of the model and suggested that further validation and ablation studies were necessary to demonstrate the effectiveness of the proposed approach. Additionally, reviewers expressed concerns about the potential loss of fine-grained linguistic details when converting text into visual tokens and the lack of interpretability analyses to understand how the model integrates image and text information. The authors did not provide sufficient responses or clarifications to these critical points during the rebuttal period. After carefully weighing these unresolved issues, it was determined that the paper does not meet the acceptance standards at this time.

---

### Decision · Program_Chairs · 2025-01-22

Reject